# Fluctuations of Physiological Variables during Conditioning of Lipizzan Fillies before Starting under Saddle

**DOI:** 10.3390/ani12070836

**Published:** 2022-03-25

**Authors:** Nina Čebulj-Kadunc, Robert Frangež, Peter Kruljc

**Affiliations:** 1Institute of Preclinical Sciences, Veterinary Faculty, University of Ljubljana, Gerbičeva 60, 1000 Ljubljana, Slovenia; nina.cebulj.kadunc@vf.uni-lj.si (N.Č.-K.); robert.frangez@vf.uni-lj.si (R.F.); 2Clinic for Breeding and Health Care of Horses, Veterinary Faculty, University of Ljubljana, Gerbičeva 60, 1000 Ljubljana, Slovenia

**Keywords:** Lipizzan breed, fillies, exercise testing, infrared thermography, physiological values

## Abstract

**Simple Summary:**

Equine exercise physiological research has led to the development of scientifically sound programs that improve the physical fitness of horses. However, there are few scientific studies on the physical responses of young horses during initial training, although they are known to be sensitive to exercise overload. Our study focused on purebred Lipizzan fillies, aged 4 years in the early stages of training. To investigate the response of the fillies to a workload achieved by lunging, heart, and respiratory rates, rectal and skin temperatures, and blood parameters, cortisol and lactate concentrations were measured. All measured values, which were within a normal range for warm-blooded horses, increased after exercise due to the increased requirements of the working muscles and heat production. Despite the increase in surface temperatures of different body regions after exercise, their distributions remained unchanged. It was highest in the front regions, followed by the rear regions, and lowest in the lower parts of the legs. Our study contributes to the knowledge of physiological processes in young horses during exercise, as well as supplements research in the field of equine exercise testing, sports physiology, and animal welfare, and provides important knowledge for the conservation and development of the Lipizzan breed.

**Abstract:**

Scientific studies on the physiological responses of young horses to workloads are limited. Therefore, the aim of our study was to determine the basal values of some cardiovascular, thermoregulatory, hematological, and biochemical parameters in 10 purebred Lipizzan fillies aged 4 years in the initial phase of training, and their responses to a graded workload, i.e., by lunging for 15 min in four exercise tests at 2-week intervals. The basal values of the measured parameters were within a range for warm-blooded horses and mostly increased after exercise in all four exercise tests. Resting heart rates were above physiological values at the baseline but decreased as the study progressed. Bilateral symmetry of body surface temperatures (BSTs) was confirmed at rest and after exercise. The highest BSTs were measured at the cranial, followed by the caudal and distal body regions. A moderate increase in cortisol and a small increase in lactate concentration indicated a low intensity of workload. The results presented contribute to the knowledge of the complex physiological processes that occur in young horses during exercise and provide a basis for further research into the field of sports physiology and welfare, as well as the conservation and development of the Lipizzan breed.

## 1. Introduction

The training of warm-blooded sport horses usually begins at the age of three years. Young horses being trained for the first time are exposed to a variety of stressors that include various anthropogenic, environmental, and social situations [1,2,3,4,5,6]. Therefore, the first training of a young horse requires the reduction of fear reactions towards the trainer and the habituation of the animal to new situations and tasks [3,6]. Adolescent domestic horses are usually kept in herds that offer them many opportunities for social interaction and free movement. These horses, at the beginning of riding training, are often separated from their groups, to be housed individually [7]. The social and environmental conditions to which a horse is exposed to can influence its ability to respond to challenging situations, alter its behavior and physiological responses, and affect its welfare. If the conditions are unfavorable, the animal’s ability to learn and train may decrease, which may also reduce the value of the horse [3,8]. Inappropriate training practices and/or management can lead to conflicting behavior, which endangers the safety of riders and grooms [9] and is a possible reason for the alarmingly high dropout rates in young horses [10], which in turn is also an underestimated welfare problem [11].

Equine exercise physiological research has led to the development of scientifically based programs [12,13,14,15], to improve the physical fitness of equine athletes [6,16]. Scientific studies on the physical responses of young horses during initial equestrian training are limited [6], although they are known to be sensitive to exercise overload [17,18]. Our previous study showed the responses of cardiovascular, respiratory, and thermoregulatory parameters to graded exercises (i.e., lunging) in adult Lipizzan horses (Lipizzans), establishing a foundation for further research in the field of equine exercise testing and sports physiology [19]. The Lipizzan is one of the oldest European horse breeds; it is considered endangered due to the small population [20]. Lipizzan stallions are highly successful in classical dressage, but the popularity of the breed is also rising in modern equestrian sports, such as dressage and combined driving [20]. The physiological responses of Lipizzans to workloads are not sufficiently understood and the standards for evaluating their fitness during training or performance are inadequate [19]. The aim of the present study was to determine the baseline values of some physiological parameters (heart rate, respiratory rate, rectal temperature, and body surface temperature) and acclimatization to a graded training load in Lipizzan fillies during the initial phase of training. We examined the course of physiological responses to graded workloads and used serum cortisol fluctuations to assess the degrees of stress to which the fillies were exposed. The results of our study will contribute toward the development of standards and protocols for monitoring fitness levels and training progress in Lipizzan horses and will help in monitoring the health status (and in assessing the well-being) of horses in general.

## 2. Materials and Methods

### 2.1. Animals

The study was conducted at the Lipizza Stud Farm (Slovenia), the oldest European stud farm that breeds Lipizzan horses (Lipizzans), which is one of the oldest cultivated breeds of horses [20]. Ten purebred Lipizzan fillies aged 4 years (all born in 2014) and with an average body weight of 455 ± 36 kg, previously selected by the stud experts for further breeding, were included in the study. They were housed together in a stable with straw bedding and were fed, *ad libitum*, hay, with free access to automatic drinkers. In addition, they received 0.5 to 1 kg of oats daily. Regarding individual history and clinical examinations, including the detection of lameness and continuous monitoring of physiological parameters, the examined fillies were considered clinically healthy throughout the study.

Handling of the studied fillies followed the traditional procedures of the stud according to the official Slovenian standardized breeding program for Lipizzans [21]. After weaning at the age of 6 months, the foals were separated by sex and integrated into herds, kept in a stable pasture system. At the age of 3.5 years, horses that were selected for further breeding were brought into stables and included in the work introduction, which lasted at least 6 months (or longer, in the case of less efficient animals). Afterward, the working test was carried out, which allowed an assessment of the character, temperament, quality of movement (free and under saddle), and general physical performance, as well as willingness and ability under the saddle and in the harness. Depending on the results of the working test, the training of the horse in question is directed towards dressage, driving in harness or tourist riding.

All procedures described were performed in accordance with established professional and ethical standards of horsemanship. Animal use (and care) was carried out in accordance with the protocol approved by the Commission for Animal Welfare of Veterinary Faculty, University of Ljubljana.

### 2.2. Test Protocol and Physical Activity

The first exercise test (ExT-1) was performed in December 2018, followed by three additional tests (ExT-2, ExT-3, and ExT-4), 14 days apart. Each exercise test was performed by lunging in an open riding arena (20 × 50 m^2^) with a sandy ground that allowed a lunging radius of 8.5 m. The leading and lunging of each filly was performed by a professional trainer, who was responsible for the fillies throughout the entire work introduction process. The exercise tests consisted of three phases with specific activities (before exercise (BEx), during exercise (Ex), and after exercise (AEx; Table 1). Each filly was brought to the riding arena 5 to 10 min before the start of the experiment and immediately fitted with the heart rate monitor. Then, the first phase of the test (BEx) began, which lasted 10 min and was used to measure baseline surface temperatures (BST) of different body regions, rectal temperature (RT), respiratory rate (RR), and heart rate (HR) in resting fillies. Blood was also drawn at this stage (BEx) for hematological, biochemical, and endocrinological analyses. In the second phase, each filly was lunged (Ex) for 15 min, gradually increasing the workload from walk to trot to canter. Each gait lasted 5 min and the reins were changed at each gait passage. In the third–last phase after the end of the exercise (AEx), immediately after the horse was stopped, all measurements and sampling were repeated (as described for BEx (Table 1)) and completed within 10 min.

### 2.3. Measurements and Equipment

Heart rate (HR) was recorded telemetrically using a heart rate monitor consisting of a transmitter (Polar Equine H3 heart rate sensor belt set, Polar Electro, Kempele, Finland) and a receiver (Polar Equine V 800, Polar Electro, Kempele, Finland) installed as previously described [19]. The Polar Flow application (Polar Electro, Kempele, Finland) was used to display and analyze the recordings. Respiratory rate was measured directly by counting the respiratory movements of the thorax and abdomen for 60 s. Rectal temperature was measured with a digital thermometer, Geratherm Flex (Geratherm Medical AG, Geschwenda, Germany). Body weight was determined with a Horse and Pony Weight Tape (Equivet, Kruuse, Langeskov, Denmark).

Body surface temperatures (BSTs) were measured within 1 to 2 min with a FLIR infrared thermal imaging camera (model E40bx, FLIR systems, Wilsonville, OR, USA) at a distance of 0.75 to 1.0 m from the left and right side of each body region (neck—*R. coli lateralis*, chest—*R. pectoralis*, back—*R. lumbalis*, croup—*R. sacralis*, buttocks—*R. femoralis*, metatarsus—*R. metatarsalis*, and metacarpus—*R. metacarpalis)*. The captured images were processed using the FLIR Tools application (FLIR system, Wilsonville, OR, USA) to determine the average BST of the selected regions. Ambient temperature and humidity (Table 2) were measured using a digital humidity meter, Testo 635 (Testo AG, Lenzkirch, Germany).

Blood was collected via jugular vein punctures with double-ended needles (Vacuette^®^, 20G x 11/2’’, Greiner Bio-one, Kremsmünster, Austria) and vacuum tubes (Vacuette, Greiner Bio-one, Kremsmünster, Austria). Blood samples for serum cortisol were collected in serum separator tubes containing clot activator (Vacuette^®^ 454,228), centrifuged for 10 min at 1300× *g* and 4 °C, and the serum was stored at −20 °C until analysis. Blood samples for the determination of plasma lactate were collected in tubes containing lithium iodoacetate and heparin (Vacuette^®^ 454,081). Tubes with the anticoagulant K_3_EDTA (Vacuette^®^ 454,222) were used to collect blood for the hematological analyses.

### 2.4. Laboratory Analyses

Hematological analyses were performed using the ADVIA 120 automated hematological analyzer (Siemens, Erlangen, Germany) immediately after samples arrived at the laboratory, 2 to 6 h after sampling. Plasma lactate was measured immediately after blood sampling using a Nova Vet StatSensor Xpress lactate meter (Nova Biomedical Corporation, Waltham, MA, USA).

Serum cortisol concentration was measured using a commercially available EIA kit (Cortisol EIA, cat. no. DEH 3388, Demeditec, Kiel, Germany) with a sensitivity of 0.1 g/L; instructions for the calculation of the intra- and inter-assay coefficients of variation (CV) were used for the estimation of the kit precision. CVs were 8.77% and 9.91% for low (3.88 ± 0.34 μg/L) and 4.15% and 8.49% for high serum cortisol levels (19.56 ± 0.81 μg/L). A microplate reader Multiskan FC (Thermo Scientific, Waltham, MA, USA) was used to measure absorbance and calculate concentrations automatically.

### 2.5. Data Analysis

Statistical analyses were performed by SigmaPlot 14.5 (Systat Software, Inc., Erkrath, Germany). One-way repeated measure ANOVAs were used to compare results measured before and after exercise for each exercise test and between exercise tests within each phase. Normality of the data distribution was assessed using a Shapiro–Wilk test and the relationship between the variables was determined using all pairwise multiple comparisons (Tukey test). Correlations between the results were examined with the calculations of Pearson product–moment correlation coefficient. To compare the skin temperatures between the regions on the left and right sides of the horses, a pairwise Student t-test was applied. Values in text are presented as means ± standard error of the mean ($\bar{x}$ ± SE). The differences are considered significant at *p* ≤ 0.05.

## 3. Results

### 3.1. Gait Speeds during Exercise

In the Lipizzan fillies, a significant increase in gait speeds during the transitions from walk to trot to canter was observed in all tests (Table 3). The slowest trot and canter were recorded in ExT-1 and the fastest in ExT-3 and ExT-4 (*p* < 0.05, respectively). For the walk, the differences between the tests were insignificant (*p* > 0.05). Trotting speed correlated positively (R = 0.9920; *p* < 0.01) with ambient temperature (for data see Table 2). Positive, although non-significant, correlations (R = 0.673, *p* > 0.05) were also observed between canter speed and ambient temperature (Table 2) and between trotting and canter speed and progression of the study (R = 0.946 and R = 0.599, respectively).

### 3.2. Rectal Temperature, Respiratory Rate, and Heart Rate

The mean rectal temperatures (RT) of the examined fillies before and after exercise in all four tests are shown in Table 4. Before exercise, the highest RT was measured in ExT-2 (*p* < 0.05 compared to ExT-4). After exercise, the highest RT was measured in ExT-1 (*p* < 0.01 compared to ExT-3). RTs increased significantly after exercise in all tests (see Table 4), except ExT-2. Rectal temperature correlated negatively (BEx: R = −0.881, *p* > 0.05; AEx: R = −0.897, *p* > 0.05) with ambient temperature (for values see Table 2). The highest respiratory rate (RR) was measured in ExT-2 before exercise and in ExT-4 after exercise, although the differences between tests were insignificant in both BEx and AEx. RR increased significantly in all tests AEx (Table 4) and correlated positively (R = 0.682 BEx, *p* > 0.05; R = 0.863 AEx, *p* > 0.05) with ambient humidity (for values see Table 2).

The heart rate (HR) gradually increased (Figure 1) from walk and trot to peak at canter (*p* < 0.001 compared to BEx in all tests) and to decrease at rest AEx (*p* < 0.001 compared to canter in all tests). The highest HR before exercise was measured in ExT-1 and the lowest in ExT-3. Significant differences between groups were found BEx with the highest HR in ExT-1 (*p* < 0.001 compared to ExT-4). In walk, trot, and canter, the differences among the tests were not significant (*p* > 0.05), although the highest heart rates for all gaits were measured in ExT-1. The heart rates BEx and AEx were negatively correlated (*p* > 0.05) with the ambient temperature (for values see Table 2) and progression of experiments.

### 3.3. Body Surface Temperatures

The differences in body surface temperatures (BSTs) between the contralateral sides of all examined body regions during all four exercise tests were not significant (*p* > 0.05). Therefore, the average temperatures considering both sides of each region were calculated and used for further analyses. The distribution of BSTs prior to physical activity (Figure 2) followed similar patterns in all four exercise tests: the highest basal BSTs were measured in the chest, neck, and buttocks; the temperatures of these regions did not differ significantly among themselves (*p* ≥ 0.05), as was also the case for back vs. croup with the intermediate temperatures and metatarsus vs. metacarpus with the lowest temperatures. The differences between the groups of regions with the highest, medium, and lowest basal temperatures within a test were significant (*p* < 0.001, *p* < 0.05, see Figure 2). The lowest BSTs were measured, BEx in ExT-1, while the highest in ExT-4 for chest (*p* < 0.01), neck (*p* < 0.01), buttocks (*p* < 0.001), metatarsus (*p* < 0.001,) and metacarpus (*p* < 0.01), and in ExT-2 for back (*p* < 0.001) and croup (*p* < 0.01), compared to ExT-1. An increase in BST was observed AEx in all four tests including all measured regions (Figure 2). Compared to BEx values, the increase in BST was significant in all four tests for chest, neck, back, croup, and buttocks (except for ExT-2). In contrast to the other regions, the increase in BST was not significant for the metacarpus and metatarsus (except for ExT-1), and for the metatarsus, an increase in BST was observed in ExT-3 (Figure 2). BST correlated positively with ambient temperature (for values see Table 2) BEx (from R = 0.251 for back to R = 0.941 for metacarpus, *p* > 0.05, respectively) and AEx (from R = 0.595 for back to R = 0.951 for metacarpus, *p* >0.05, respectively).

The areas that warmed up the most after exercise in all four tests were the neck, followed by the chest, back, croup, and buttocks (Figure 3). The areas that warmed the least were the metacarpus and metatarsus; in ExT-3, the metacarpus actually cooled after exercise (Table 5). The differences in heating between regions were not significant between tests (*p* > 0.05), with the exception of the metatarsus (*p* < 0.05). In contrast, when comparing different regions within a single test, significant differences in heating were confirmed for ExT-2 (*p* < 0.001), ExT-3, and ExT-4 (*p* < 0.0001, respectively). Significant differences were also found in the heating of the warmest (neck, chest) and coldest regions (metacarpus, metatarsus; see Table 5).

### 3.4. Biochemistry and Hematology

In all exercise tests (Table 6), cortisol concentrations (CORT) BEx were similar (*p* > 0.05). A significant increase in CORT was observed in all four exercise tests AEx compared to BEx values (*p* < 0.001 for ExT-1, EXT-3 and ExT-4, *p* 0.01 for ExT-2; Table 6). The highest increase was measured in ExT-3 and the lowest in ExT-4 (97.51 ± 4.39 nmol/L and 71.96 ± 4.32 nmol/L, respectively; *p* < 0.001). Positive correlations were found between the CORT and the respiratory rate BEx (R = 0.950, *p* < 0.05) and AEx (R = 0.958, *p* < 0.05).

The highest lactate concentration (LAC) was found BEx in ExT-1 (*p* ≥ 0.05), while it was significantly lower in ExT-2, ExT-3 and ExT-4 (*p* < 0.05, respectively; Table 6). A non-significant increase in LAC was observed AEx in ExT-1, 2 and 4 compared to BEx values; an opposite response was observed in ExT-3 (Table 6). LAC correlated negatively (BEx: R = −0.708; AEx: R = −0.782, *p* > 0.05, respectively) with ambient temperature (for values see Table 2). A positive correlation was observed between LAC and HR both before and after exercise (R = 0.800 and 0.882, *p* > 0.05 and = 0.118, respectively) and among LAC and RBC, HGB, and HCT (R = 0.639 to 0.992), which was significant for RBC (BEx: *p* < 0.05, AEx: *p* < 0.01) and HGB BEx (*p* < 0.05).

The hematological values (red blood cell (RBC) and white blood cell (WBC)) counts, hemoglobin concentration (HGB), and hematocrit (HCT) of the examined fillies BEx and AEx in tests 1 to 4 are shown in Table 7. Differences between the samplings were found for several measured parameters. The values of all presented hematological parameters increased in AEx compared to BEx values in all tests, except for ExT-1, where the opposite situation—a decrease in values—was observed (Table 7). Negative correlations were found between ambient temperatures (for values see Table 2) and RBC, HGB, and HCT, not significant BEx and AEx.

## 4. Discussion

The aim of this study was to measure the baseline values of some physiological (heart and respiratory rate, rectal, and body surface temperature), serum biochemical (cortisol and lactate concentrations) and hematological parameters (red and white blood cell counts, hemoglobin concentration, and hematocrit) of Lipizzan fillies, their changes with graded workloads and their acclimatization to workload over a longer period of time. In addition, we were also interested in the changes in the measured values during stress generated by different procedures during the initial training of young horses. Responses to graded workloads by lunging were investigated in four exercise tests (ExT-1 to ExT-4) two weeks apart; measurements of values of interest were taken before and after exercise (BEx and AEx). We believe that this is the first study dealing with exercise testing of fillies belonging to the Lipizzan breed.

### 4.1. Gait Speeds

The changes in gait speeds of the tested Lipizzan fillies followed similar patterns in all four exercise tests, as expected, with a significant increase in speed at the transition from walk to trot to canter. The walk speed was in line with the reference values for horses, the trot speed was in the lower range of the reference values for this gait and the canter speed was below the reference values [12,22]. Our previous study [19] on adult Lipizzan horses during lunging likewise demonstrated slower gaits than in horses during riding [23,24]. This could be a characteristic of Lipizzans or caused by the movement in a circle during lunging, which could have caused a less efficient combination of stride frequency and length than with a straightforward movement [12]. Another more likely explanation for the slow trot and canter of the Lipizzan fillies studied is the way they were handled during the exercise test. Namely, the trainers required the fillies to change from walk to trot and finally to canter in 5-min intervals. During this process, all physical coercion to speed up the pace was avoided, apart from voice, changing the tension of the lunge rope, and pointing the whip at the horse’s hindquarters. Trot and canter speeds increased simultaneously as the study progressed, indicating a gradual improvement in physical fitness [6,16].

### 4.2. Rectal Temperature (RT), Respiratory Rate (RR), and Heart Rate (HR)

Mean RTs measured Lipizzan fillies BEx were in a range previously reported for warm-blooded horses [25,26]. RTs increased after exercise, reflecting accelerated heat production during physical activity in all exercise tests [27]. Although a negative correlation was found between the RT and ambient temperature, which increased steadily throughout the study, differences in RTs were only BEx between ExT-2 and ExT-4 and AEx between ExT-1 and ExT-3. As the differences between tests were not significant, a causal relationship between these two variables seems unlikely.

Mean RRs measured in fillies BEx ranged from 10 to 18 breaths per minute, which is a normal range for warm-blooded horses [26,28]. The pattern of RR changes was similar in all exercise tests, indicating a significant increase in AEx values, reflecting the physiological responses of the horse to increased oxygen consumption during exercise [28,29], but also contributing to evaporative heat loss from the respiratory tract [25]. The role of air humidity in thermoregulation was indicated by a positive but non-significant correlation between the respiratory rates of Lipizzan fillies studied and the humidity before and after the exercise tests [25,26,27,28,29].

The reaction of HR to exercise is a function of work intensity and reflects the metabolic performance of horses. It is under the influence of various external and internal factors, including environmental conditions, fitness and health status of the horse [14,17,19,30]. The mean HRs in Lipizzan fillies BEx were above the physiological range for warm-blooded horses [12,19,26,28,31], with the highest value measured at ExT-1. This indicates the excitement of the fillies caused by leaving the box, installing the HR measuring devices and meeting unfamiliar people before the exercise test [12,13,14,26,28,31]. A similar psychogenic response of HR to stress was also found in racehorses just before the start of a competition and in warmblood horses during handling, saddling or harnessing before training. On the other hand, the presence of the horse’s handler or a calm companion reduced HR in stressful situations. The highest HR value before the exercise (while the fillies were standing still) was measured at ExT-1, followed by a significant decline in the further three tests, indicating a gradual habituation of the fillies to the events and procedures BEx [3,6,12,13,14,17,26,28,31].

During physical activity, the HR changes in Lipizzan fillies followed a similar pattern and reached similar values in all exercise tests, namely a gradual increase with the enhancement of load following transition to faster gaits. Heart rates were in the ranges reported for other horse breeds in comparable studies and did not reach the maximum HR, which can exceed 200 beats/min in horses [12,19,23,26,29,30,31,32]. Contrary to our expectations, no differences in HRs were found between trot and canter, as similar speeds were measured at both gaits. Within ten minutes after the exercise, HR values decreased significantly in all four tests and reached HRs lower than BEx, indicating a normal physiological response and a good fitness status of studied fillies [12,19,23,26,29,30,31,32].

### 4.3. Body Surface Temperature (BST)

At rest, the metabolic activity of the muscles generates a constant amount of heat, which increases proportionally to the workload during exercise. To sustain body temperature within physiological limits, thermoregulatory mechanisms are activated, including increased blood flow through the skin capillaries, resulting in an increase in BST [15,25,27]. Regarding BSTs, horses are bilaterally symmetrical [19,25,33], which was also confirmed in this study in Lipizzan fillies for all exercise tests, indicating a balanced muscle work and an appropriate track surface during exercise. Therefore, the average BSTs were calculated and further proceeded considering both sides of each region of interest [19].

The warmest regions in Lipizzan fillies at rest, BEx, were the chest, neck, and buttocks, the coldest were the metacarpus and the metatarsus; intermediate BSTs were measured on the back and croup. All these values were well below the values of around 30 °C reported in other studies [15,25,28,33] or previously measured in adult Lipizzan horses [19], indicating the influence of different ambient temperatures on BST in horses. Average ambient temperatures during our study, ranging from 6.7 ± 1.1 °C (ExT-1) to 11.1 ± 0.5 °C (ExT-4), were lower than in previous studies in horses [15,19,25,33,34], which may be the reason for the lower BSCs in the Lipizzan fillies studied. Increased heat loss, proportional to the decrease in ambient temperature, results mainly from the decrease in peripheral blood flow due to the direction of blood to the internal organs, the thermal insulation of skin and coat and the amount of subcutaneous fat [28,35], but may also depend on the type of horse [35]. An inverse correlation between the gradual decrease in ambient temperature towards the lower critical temperature (5 °C) and the rate of non-evaporative heat loss in horses is almost linear [36] and has been confirmed by variations in limb surface temperatures that are directly related to changes in ambient temperature between 5 and 25 °C [28]. A positive but insignificant correlation between BSTs and ambient temperatures was also found with progression of our study. We believe that a gradual increase in ambient temperatures was not high enough to significantly affect the BSCs of the Lipizzan fillies studied.

Several studies on horses describe changes and distributions of surface temperatures on different body regions before and after exercises and confirm the influence of exercise on skin heating [15,25,33,34,37,38]. Body surface temperatures also increased in the Lipizzan fillies studied after exercise in all exercise tests. The areas with the highest BST in all four tests, AEx, was the neck, followed by the chest [25,38], suggesting that the largest clusters of maximum temperatures lie over the anatomically largest surfaces, which play a significant role in thermoregulatory function. In the Lipizzan fillies studied, medium BSTs were measured on the back, croup and buttocks, and the areas that warmed least were the metacarpus and metatarsus; in ExT-3, the metacarpus actually cooled after exercise, in contrast to a study by Simon et al. (2006) [33], who reported an increase in BSTs on the distal parts of the limbs after exercise. Different BSTs of various regions within individual tests of Lipizzan fillies indicate different muscle work of these regions, with the cranial parts of the body carrying a greater load in supporting and moving the body than the caudal parts. Increased workload accelerates metabolic rate and oxygen demand, which in turn increases muscle blood flow. These reactions eventually lead to increased blood flow to the skin, which aims to release excessive heat into the environment [15,16,25].

The highest heating (temperature difference between the BEx and AEx values) of the BSTs in Lipizzan fillies after all four exercise tests was observed for the neck, followed by the chest. An intermediate warming was observed for the back, croup and buttocks. The regions that warmed the least were the metatarsus and metacarpus: the maximum warming was registered in test 1 (3.4 °C for the metatarsus and 2.4 °C for the metacarpus). This difference then gradually decreased and reached 0.3 °C in ExT-4 for both regions. In ExT-3, the metatarsus even cooled by 0.4 °C. The temperature difference of various body regions BEx and AEx in the examined Lipizzan fillies was in the ranges reported for horses under comparable workloads [33], but higher than in horses after jumping competitions [25] or Felin ponies [15] exposed to mild exercise on the treadmill. Our results suggest that heating (increase in body surface temperature after exercise) is independent of ambient temperature, and also confirm the different contributions of the different regions to thermoregulation, and the different loads they carry in supporting and moving the body [15,25,33,34,37,38].

### 4.4. Biochemistry and Hematology

Changes in serum cortisol concentration (CORT) are objective and useful indicators for assessing stress in horses [39]. Horses may be exposed to various potentially stressful situations, including physical training, equestrian competitions and veterinary examinations [6], which was also the case in our study. The average CORT concentrations in the Lipizzan fillies studied were within the ranges reported for resting horses [39,40] or slightly below [1,22,41] in all four tests, BEx. An increase in plasma or salivary CORT is noted in horses after various types of exercise [6,41,42]. A similar increase in serum CORT was observed in our study during each exercise test comparing AEx and BEx values. This clearly demonstrates a response of the hypothalamic-pituitary-adrenal axis to the increased metabolic demands in exercising horses [41]. Due to the design of our study, we were not able to distinguish between the responses of the horses studied to exercise or other possible stressors, such as contact with unfamiliar people or environments. Although the correlations between circulation and saliva CORT are well known [6,7], it is impossible to compare the values measured with different methods. Nevertheless, salivary cortisol concentrations achieved in response to exercise were significantly lower than in horses during road transport [6].

Lactate is constantly produced in working muscles and remains low until its production exceeds excretion. The excess lactate is excreted to allow the other organs to recycle their energy resources by entering the Cori cycle [43,44]. In addition to clinical use, lactate can also be measured during exercise to monitor the fitness and performance of sport horses [17,44,45]. Here, lactate accumulation is used as an indicator of fatigue after anaerobic exercise and indicates that fatigue is reduced by the elimination of lactate [46]. The normal lactate concentration in the blood of a resting horse is 1–2 mmol/L [17,43,44,46]. Similar values were also measured in the Lipizzaner fillies, BEx, when they stood at rest. The lactate concentration in the blood of horses increases exponentially with increasing speed and can reach 25 to 30 mmol/L after peak speeds or physical exertion [23,45]. Contrary to our expectations, the increase in lactate concentration in the tested Lipizzan AEx did not reach the limit of the anaerobic threshold at 4.00 mmol/L [18]. In tests 1, 2, and 4, only a slight increase in values compared to baseline and even a decrease in ExT-3 was observed. These low lactate concentrations indicate a low production and rapid degradation of lactate in the examined fillies, which is probably due to the low intensity of the exercise tests and proves a good physical condition in which most of the energy for muscle work is produced by aerobic metabolism [45].

The values of hematological parameters measured in Lipizzan fillies before and after exercise were within the ranges for horses [15,16,47] and were in agreement with the values measured in our previous studies on Lipizzans [48,49]. Increases in WBC, RBC, HGB, and HCT were found in response to exercise, which could be due to contraction of the spleen at the onset of exercise [15,50]. Another reason for these changes could be mild dehydration due to sweating [15,16,17], but this is less likely as the study was conducted at low ambient temperatures. Nevertheless, the changes in the hematological parameters AEx were small, indicating a low physical burden to which the Lipizzan fillies studied were exposed.

## 5. Conclusions

In conclusion, the results of our study show the physiological responses of Lipizzan fillies to graded exercise during four consecutive exercise tests at 2-week intervals. The gait speeds of the tested fillies increased during the transition from walk to trot and canter in all four exercise tests. The walk speed was in line while the trot and canter speeds were below the reference values for horses, but increased during the study, indicating a gradual improvement in physical fitness. Rectal temperature (RT) and respiratory rate (RR) in Lipizzan fillies at rest before exercise were within normal values for horses and increased during exercise in all four tests due to increased metabolic demands. The mean heart rates (HR) of Lipizzan fillies before exercise were above the physiological range for warm-blooded horses, with the highest value measured at the beginning of the study, followed by a significant decrease in the other three tests, indicating a gradual acclimatization of the fillies to the events and procedures before exercise. During physical activity, the HR gradually increased with the increase in load during the transition to faster gaits but remained within the reported ranges for exercising horses. Within ten minutes after exercise, HR values dropped below basal values in all exercise tests, indicating a normal physiological response and a good fitness status of the examined fillies. The highest body surface temperatures (BSTs) were measured in Lipizzan fillies at rest before exercise at the chest, neck and buttocks, followed by medium BSTs at the back and croup and the lowest at the metacarpus and metatarsus. After exercise, the BSTs of all regions of interest increased in all exercise tests, with a similar temperature distribution as before exercise, namely highest in the cranial, followed by the caudal and lowest in the distal regions. All values were lower than in horses tested at higher ambient temperatures, indicating the influence of ambient temperatures on BSTs. Surprisingly, the temperature differences in the Lipizzan fillies tested reached similar values as in horses trained at higher ambient temperatures, suggesting that the warming range might depend on the heat production during training and not on the ambient temperature. The average CORT concentrations in examined fillies were within the ranges reported for resting horses and increased after exercise in all tests, as would be expected in exercising horses. The lactate concentration measured in the Lipizzaner fillies at rest before exercise was within the normal range for resting horses and did not reach the limit of the anaerobic threshold of 4.00 mmol/L after training, indicating a low intensity of exercise and a good physical condition of the examined fillies. The values of hematological parameters were within the usual ranges for horses before and after exercise, although WBC, RBC, HGB, and HCT increased slightly in response to exercise, indicating a low physical stress to which the examined fillies were exposed.

The variations in the measured parameters over the course of the study indicate a gradual habituation of the fillies to the work schedule, a gradual improvement in their physical conditions, and an adaptation to environmental changes. This study contributes to the knowledge of the complex physiological processes that occur during exercise and provides a basis for further research in the field of exercise testing, sports medicine, and equine welfare. The results presented can also be seen as an important contribution to the preservation and development of the Lipizzan breed.

## Figures and Tables

**Figure 1 animals-12-00836-f001:**
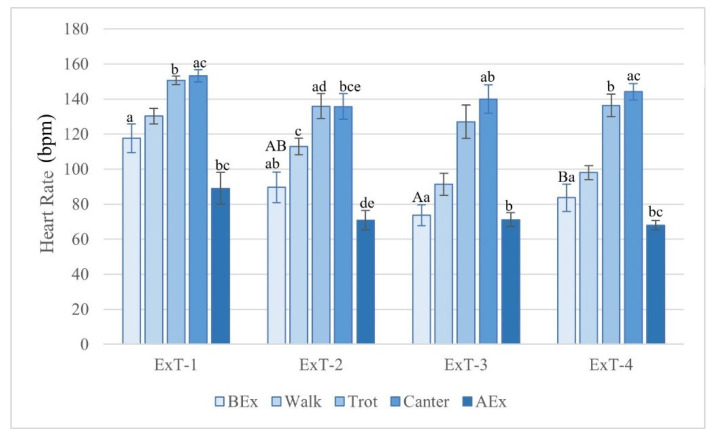
Heart rates measured in different phases of exercise tests 1 to 4. Legend: bpm—beats per minute; ExT-1 to ExT-4—exercise tests 1 to 4; BEx—before exercise; AEx—after exercise. ^a,b,c,d,e^ *p* < 0.001 (for values with the same label within a test). ^A, B^ *p* < 0.001 (for values with the same label within the same phase of different tests).

**Figure 2 animals-12-00836-f002:**
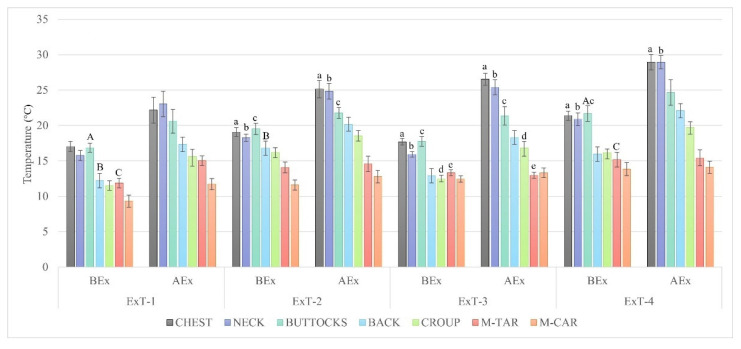
Body surface temperatures of different regions before and after exercise (BEx and AEx) in tests 1 to 4 (ExT-1 to ExT-4). M-TAR—metatarsus; M-CAR—metacarpus; ^a,b,c,d,e^ *p* < 0.001 within a test (before vs after exercise); ^A,B,C^ *p* < 0.001 between tests.

**Figure 3 animals-12-00836-f003:**
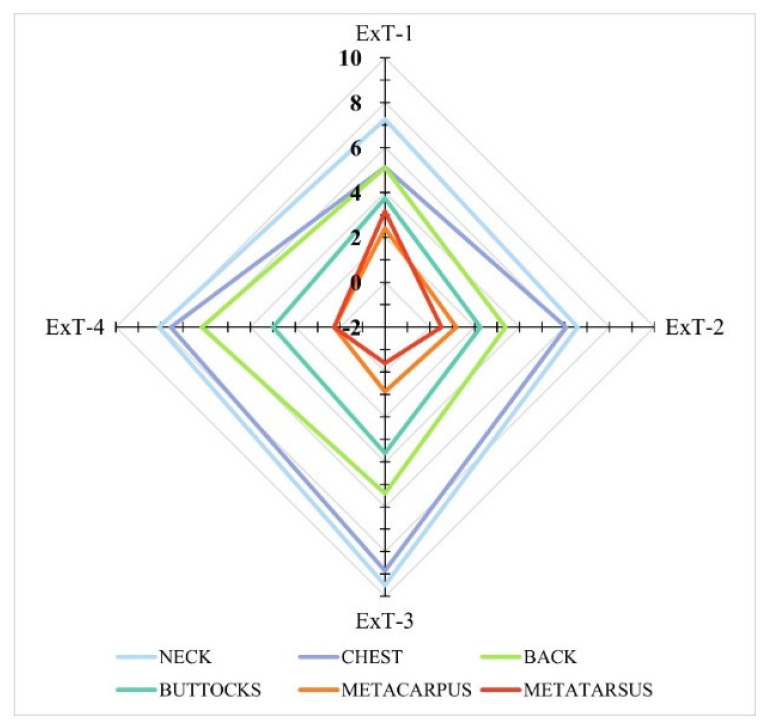
Graphical presentation of heating (temperature differences of body surface regions before and after exercise; scaling: absolute temperatures (°C)). ExT-1 to ExT-4—exercise tests 1 to 4.

**Table 1 animals-12-00836-t001:** Phases of each exercise test (duration of activity, its type, recordings, and samplings are given for each phase of the test).

Phase of the Test	Duration of Activity (min)	Activity	Recordings and Samplings *
BEx	10	Rest	BST, RT, RR, HR, VP
Ex	5	Lunging (walk)	HR, Va
	5	Lunging (trot)	HR, Va
	5	Lunging (canter)	HR, Va
AEx	10	Rest	BST, RT, RR, HR, VP

Legend: Bex—before exercise; Ex—exercise; AEx—after exercise; BST—body surface temperature; RT–rectal temperature; RR—respiratory rate; HR—heart rate; VP—blood sampling; Va—average speed. * Air temperature and pressure were measured throughout the testing (see Table 2 for details).

**Table 2 animals-12-00836-t002:** Mean air temperature and humidity during exercise tests 1 to 4 (ExT-1 to ExT-4).

Parameter	ExT-1	ExT-2	ExT-3	ExT-4
Temperature (°C)	6.7 ± 1.1	8.0 ± 0.5	10.6 ± 0.4	11.1 ± 0.5
Humidity (%)	46.9 ± 5.1	75.4 ± 2.2	39.3 ± 1.9	65.4 ± 1.5

**Table 3 animals-12-00836-t003:** Mean gait speed achieved during exercise in tests 1 to 4 (ExT-1 to ExT-4).

Phase of the Test	Speed (km/h)			
	ExT-1	ExT-2	ExT-3	ExT-4
Walk	5.2 ± 0.4 ^a^	4.7 ± 0.2 ^b,c^	4.8 ± 0.3 ^d,e^	5.2 ± 0.2 ^b,d^
Trot	6.7 ± 0.6 ^A,B^	7.8 ± 0.5 ^b^	9.3 ± 0.5 ^d,A^	9.3 ± 0.9 ^b,B^
Canter	8.9 ± 1.2 ^a^	6.9 ± 0.7 ^c,A^	10.6 ± 0.6 ^e,A^	10.0 ± 0.7 ^d^

^a^ *p* < 0.05; ^b, c^ *p* < 0.01; ^d, e^ *p* < 0.001 for values in a column. ^A, B^ *p* < 0.05; for values in a row.

**Table 4 animals-12-00836-t004:** Rectal temperature (RT) and respiratory rate (RR) in Lipizzan fillies before and after exercise (BEx and AEx) in tests 1 to 4 (ExT-1 to ExT-4).

Parameter	Phase of the Test	ExT-1	ExT-2	ExT-3	ExT-4
RT (°C)	BEx	37.6 ± 0.1 ^a^	37.8 ± 0.2 ^A^	37.4 ± 0.0 ^b^	37.3 ± 0.1 ^a,A^
	AEx	38.5 ± 0.1 ^a,B^	38.2 ± 0.1	38.0 ± 0.1 ^b,B^	38.0 ± 0.1 ^a^
RR (/min)	BEx	17.5 ± 1.3 ^b^	25.0 ± 2.6 ^c^	16.8 ± 1.2 ^b^	16.1 ± 1.3 ^a^
	AEx	39.6 ± 5.6 ^b^	44.0 ± 6.5 ^c^	38.2 ± 3.6 ^b^	46.6 ± 7.3 ^a^

Legend: ^a^
*p* < 0.001; ^b^
*p* < 0.01; ^c^
*p* < 0.05 for values in a column; ^A^
*p* < 0.05; ^B^
*p* < 0.01 for values in a row.

**Table 5 animals-12-00836-t005:** Heating of the examined regions after exercise (differences between absolute body surface temperatures before and after exercise in tests 1 to 4 (ExT-1 to ExT-4)).

Body Region	Temperature Difference (°C)			
	ExT-1	ExT-2	ExT-3	ExT-4
Neck	7.3 ± 2.3	6.7 ± 1.2 ^b,d^	9.5 ± 0.9 ^a,b^	8,1 ± 1.5 ^c,d^
Chest	5.1 ± 1.8	6.1 ± 1.5 ^a,^^c^	8,8 ± 0.9 ^c,d^	7.6 ± 1.3 ^a,b^
Back	5.1 ± 1.7	3.4 ± 0.9	5.4 ± 1.2	6.2 ± 1.7 ^e,f^
Croup	4.1 ± 1.7	2.4 ± 0.8	4.4 ± 1.0	3.5 ± 1.1
Buttocks	3.8 ± 1.5	2.3 ± 1.0	3.6 ± 0.9	3.0 ± 1.0
Mecarpus	2.4 ± 1.0	1.2 ± 0.6 ^d,c^	0.9 ± 0.4 ^b,d^	0.3 ± 0.9 ^b,d,f^
Metarsus	3.2 ± 1.0	0.5 ± 0.7 ^a,b^	−0.4 ± 0.7 ^a,c^	0.3 ± 0.8 ^a,c,e^

Legend: ^a^ *p* < 0.001, ^b, c^ *p* < 0.01, ^d^ *p* < 0.05 for ExT-2; ^a, b, c, d^ *p* < 0.05 for ExT-3; ^a, b^ *p* < 0.001 ^c, d^ *p* < 0.01; ^e, f^ *p* < 0.05 for ExT-4.

**Table 6 animals-12-00836-t006:** Serum cortisol and blood lactate concentrations in Lipizzan fillies before and after exercise (BEx and AEx) in tests 1 to 4 (ExT-1 to ExT-4).

Parameter	Phase of the Test	ExT-1	ExT-2	ExT-3	ExT-4
Cortisol (nmol/L)	BEx	54.60 ± 3.35 ^a^	63.26 ± 5.55 ^b^	54.05 ± 3.92 ^c^	49.05 ± 4.84 ^d^
	AEx	87.89 ± 4.66 ^a^	82.81 ± 5.47 ^b^	97.51 ± 4.39 ^c, A^	71.96 ± 4.32 ^d, A^
Lactate (mmol/L)	BEx	1.0 ± 0.2	0.6 ± 0.1	0.7 ± 0.1	0.6 ± 0.1
	AEx	1.8 ± 0.7	0.8 ± 0.1	0.6 ± 0.9	0.9 ± 0.2

Legend: ^a, c, d,^ *p* < 0.001; ^b^ *p* < 0.01 for values with the same label in a column; ^A^ *p* < 0.001 for values with the same label in a row.

**Table 7 animals-12-00836-t007:** Hematological values of measured blood parameters at rest and before and after exercise (BEx and AEx) in tests 1 to 4 (ExT-1 to ExT-4).

Parameter (Units)	Phase of Test	ExT-1	ExT-2	ExT-3	ExT-4
WBC (×10^9^/L)	BEx	9.4 ± 0.2 ^A^	7.3 ± 0.6 ^A, B, G^	9.3 ± 0.3 ^B^	8.8 ± 0.2^G^
	AEx	9.2 ± 0.2 ^H^	7.8 ± 0.6 ^C, F, H^	9.7 ± 0.3 ^C^	9.3 ± 0.2 ^F^
RBC (×10^12^/L)	BEx	10.2 ± 0.2 ^A, B, C, a^	8.6 ± 0.2 ^A, a^	8.8 ± 0.2 ^B, a^	8.6 ± 0.2 ^C, a^
	AEx	9.8 ± 0.1 ^G, a^	8.9 ± 0.3 ^a^	9.1 ± 0.2 ^a^	8.6 ± 0.9 ^G, a^
HGB (g/L)	BEx	166.9 ± 3.6 ^A, B, C^	140.8 ± 2,9 ^A^	141.5 ± 2.8 ^B^	138.9 ± 3.0 ^C^
	AEx	161.3 ± 1.6 ^D, E^	146.0 ± 4.4 ^D^	146.3 ± 2.9 ^E^	154.3 ± 2.7
HCT (L/L)	BEx	0.46 ± 0.01 ^A, B, C^	0.38 ± 0.01 ^A^	0.38 ± 0.01 ^B^	0.38 ± 0.01 ^C, a^
	AEx	0.45 ± 0.01 ^D, E^	0.40 ± 0.01 ^E^	0.39 ± 0.01 ^D^	0.42 ± 0.01 ^a^

Legend: WBC—white blood cell count; RBC—red blood cell count; HGB—hemoglobin concentration; HCT—hematocrit. ^A, B, C, D, E^ *p* < 0.001; ^F, G, H^ *p* < 0.01 for values with the same label in a row. ^a^ *p* < 0.05 for values of the same parameter BEx and AEx.

## Data Availability

The original results are available from the corresponding author.

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
