# Peer review of "Fluctuations of Physiological Variables during Conditioning of Lipizzan Fillies before Starting under Saddle"

_animals, 2022, doi:10.3390/ani12070836_

Round 1

Reviewer 1 Report

the authors made an interesting study trying to assess the basal values of few variables aiming to highlight a response to Lipizzan fillies to training. The maintenance of biodiversity and the importance of such ancient breeds most of which are endangered are the main strengths of the paper. although the work design is appropriate, something can be improved:

-line 56-57: check for character dimension

  • line 86-87: this seems to me not necessary data
  • line 115 : was the trainer the same for all the fillies?
  • line 144: were body region temperature measured always following the same order of region? how long would the temperature measurement take from the first to the last region?
  •  line 162: how long after sampling, the hematology was run?
  • line 179: could you please mention in the discussion of what interest is to compare left to right side temperature? I don't get the reason
  • line 246: if the time required to make all the measurements were too long and if they were taken always in the same order, I would suppose that some regions (the last to be measured) could have cooled down. can you please provide more information about this type of measure? If ambient temperature and humidity were registered can you please show the data beside the other variable? the explanation of how the other variables link to ambient temperature and humidity is a little confused.
  • line 389-392 and line 418-422 seem to contradict each other. please clarify

Author Response

Enclosed please find our explanations!

With regards!

Reviewer 2 Report

Generally, the MS is very interesting and fully in scope of the Special Issue.

There are some imperfections and/or questions that authors should revise:

SIMPLE SUMMARY

  • Clearly give the exercise test (L18)
  • Give most significant results and highlight novelty of this study

ABSTRACT

  • Give age of the animals (L26)
  • Clearly give all significant results and outcomes

INTRODUCTION

  • Rewrite and simplify the section (L57-62)

MATERIAL AND METHODS

  • Minor error
  • Give clearly – there are 3 Ex groups in Table 1, but 4 in this part (L112-133)

RESULTS

  • Have also authors analyzed other biochemical parameters?

DISCUSSION

  • Well prepared

CONCLUSION

  • Give clearly the most important results and novelty of the study

TITLE

  • The term “certain” should be considered

Author Response

(The authors gave the same response as above.)

Round 2

Reviewer 1 Report

The authors replied to all comments.
I believe that the paper is better presented. 
Thank you